# Nickel-catalysed retro-hydroamidocarbonylation of aliphatic amides to olefins

Jiefeng Hu[1], Minyan Wang[1], Xinghui Pu[1] & Zhuangzhi Shi[1]

Amide and olefins are important synthetic intermediates with complementary reactivity which play a key role in the construction of natural products, pharmaceuticals and manmade materials. Converting the normally highly stable aliphatic amides into olefins directly is a challenging task. Here we show that a Ni/NHC-catalytic system has been established for decarbonylative elimination of aliphatic amides to generate various olefins via C–N and C–C bond cleavage. This study not only overcomes the acyl C–N bond activation in aliphatic amides, but also encompasses distinct chemical advances on a new type of elimination reaction called retro-hydroamidocarbonylation. This transformation shows good functional group compatibility and can serve as a powerful synthetic tool for late-stage olefination of amide groups in complex compounds.

[1] State Key Laboratory of Coordination Chemistry, School of Chemistry and Chemical Engineering, Nanjing University, Nanjing 210093, China. Correspondence and requests for materials should be addressed to Z.S. (email: shiz@nju.edu.cn).

Aliphatic amide skeletons are key building blocks in a broad range of natural and artificial compounds such as proteins and nylons. Resonance stabilization of the amide bond confers stable characteristic in such compounds. Although amide C–N bonds can readily be cleaved by enzymes (Fig. 1a)[1], transformations that allow direct functionalization of amide C–N bonds remains quite difficult. (Commonly used methods to cleave amide C–N bonds include the reductive conversion of amides to aldehydes using Schwartz's reagent and the displacement of Weinreb's N-OMe-N-Me amides with organometallic reagents en route to ketones[2].) Elegant work by the groups of Garg and Houk has shown that nickel catalysis is effective to promote the conversion of amide derivatives to esters by acyl C–N bond activation (Fig. 1b)[3–6]. However, all these reported methods are limited to aryl and heteroaryl amide substrates[7–9]. Nevertheless, the development of catalytic system for the efficient transformation of amides derived from aliphatic carboxylic acids is still in high demand.

The alkene moiety is vital in many broad synthetic applications including Heck reaction, electrophilic addition and olefin metathesis reaction. The ability to interconvert other functional groups with alkene is significant in synthetic chemistry[10]. For instance, as the reverse reaction of hydrohalogenation[11], regioselective dehydrohalogenations of alkyl bromides have been achieved in the presence of cobalt[12] and palladium[13] catalyst. Hydroformylation is an important homogeneously catalysed industrial process for the production of aldehydes from olefins[14,15]. In 2015, Dong and coworkers have disclosed rhodium-catalysed dehydroformylation of aldehyde substrates to olefins via transfer hydroformylation (Fig. 1c)[16,17]. Most recently, Morandi et al. have also achieved a remarkable interconversion between alkyl nitriles and alkenes by nickel-catalysed transfer hydrocyanation reactions (Fig. 1d)[18]. Inspired by these precedent results, a pathway involving decarbonylative elimination of aliphatic amides to olefin[19] developed as the reverse conversion of hydroamidocarbonylation process[20,21] seems meaningful to expand the synthetic utility (Fig. 1e).

To achieve this transformation, several difficulties need to be considered: (1) this transformation is to cleave a stable C–N bond while forming an easily transformable carbon–carbon double bond; (2) the C–N bond scission has a high activation energy and its selective cleavage in the presence of C–CN, C–O and C–F bonds is challenging; (3) the olefination products have a tendency to undergo decarbonylative Heck reaction with unconsumed amides[22]. Herein we disclose the first case of a retro-hydroamidocarbonylation process by chemoselective activation of aliphatic amide C–N bonds using nickel-catalysis[23–29].

## Results

**Reaction design**. We have recently reported nickel-catalysed decarbonylative cross-coupling of aryl amides to build C–B bonds[30,31]. As the key intermediate, the structure of the acyl nickel complex **B** (Ar = 4-methyl-1-naphthyl) was first confirmed by X-ray analysis, which displayed the square planar geometry with two carbene (ICy: 1,3-dicyclohexylimidazol-2-ylidene) ligands mutually in trans position to each other. Furthermore, the decarbonylative product **C** was also confirmed by X-ray analysis. These findings uncovered key mechanistic features of the acyl C–N bond activation process (Fig. 2, left cycle). We proposed a similar acyl nickel species **B'** as the key intermediate for further transformation in aliphatic amides. The intermediate **B'**, if formed, would undergo decarbonylation and a subsequent β-H elimination process would generate olefination products (Fig. 2, right cycle). Thus, an efficient retro-hydroamidocarbonylation process of aliphatic amides is theoretically possible.

**Investigation of reaction conditions**. Initial studies involved the evaluation of decarbonylative elimination of substrate **1** by acyl C–N bond activation (Table 1). This compound was chosen as the model substrate because of its structure, containing both ester[32–39] and amide group, and selective activation of amide C–N bond was very meaningful. In the presence of 10 mol% Ni(COD)₂, 20 mol% ICy·HCl, 20 mol% NaO$^t$Bu and 3.0 equiv K₃PO₄, at 130 °C under an argon atmosphere in toluene, we indeed observed the desired product **2** in 33% yield after 36 h in GC-MS (Table 1, entry 1). We further studied the performance of other NHC ligands such as IMes·HCl (Table 1, entry 2) and IPr·HCl (Table 1, entry 3) under these conditions and found that ICy·HCl promoted a dramatic reactivity for this transformation. It is noted that K₂CO₃ was slightly better than K₃PO₄ (Table 1, entry 4), and 44% yield of the desired product was observed by employing KOAc as the base (Table 1, entry 5). Under these conditions, changing the solvent to cyclohexane provided 51% yield of 2 (Table 1, entry 6). Using a binary solvent system, toluene and cyclohexane (v/v = 1:2) resulted in 58% yield (Table 1, entry 7). Interestingly, more improvement could be achieved employing ICy as the ligand (Table 1, entry 8). To our delight, the utilization of 0.5 equiv Mg(OAc)₂ as an additive exhibited higher reactivity and afforded

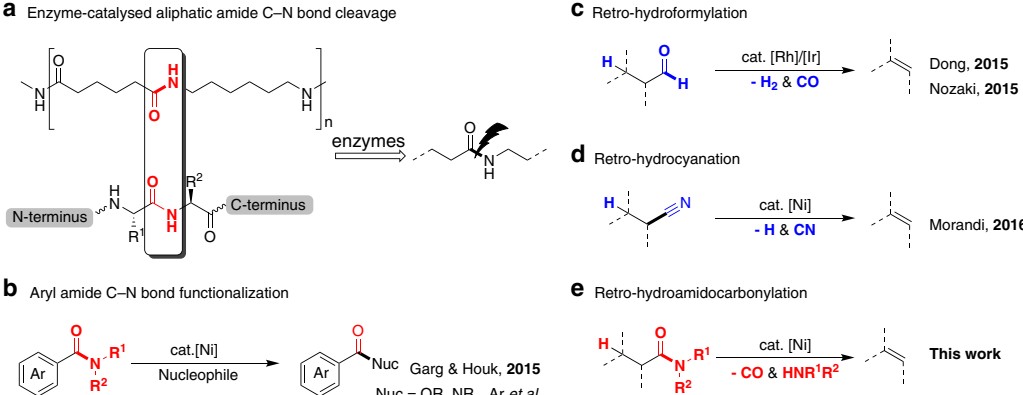

**Figure 1 | Development of a retro-hydroamidocarbonylation protocol.** (**a**) Biodegradation of nylon 66 and hydrolysis of protein. (**b**) Ni-catalysed activation of aryl amide C–N bonds. (**c**) Rh and Ir-catalysed decarbonylative elimination of aldehydes to olefins. (**d**) Ni-catalysed transformation of nitriles to olefins. (**e**) Ni-catalysed decarbonylative elimination of aliphatic amides to olefins.

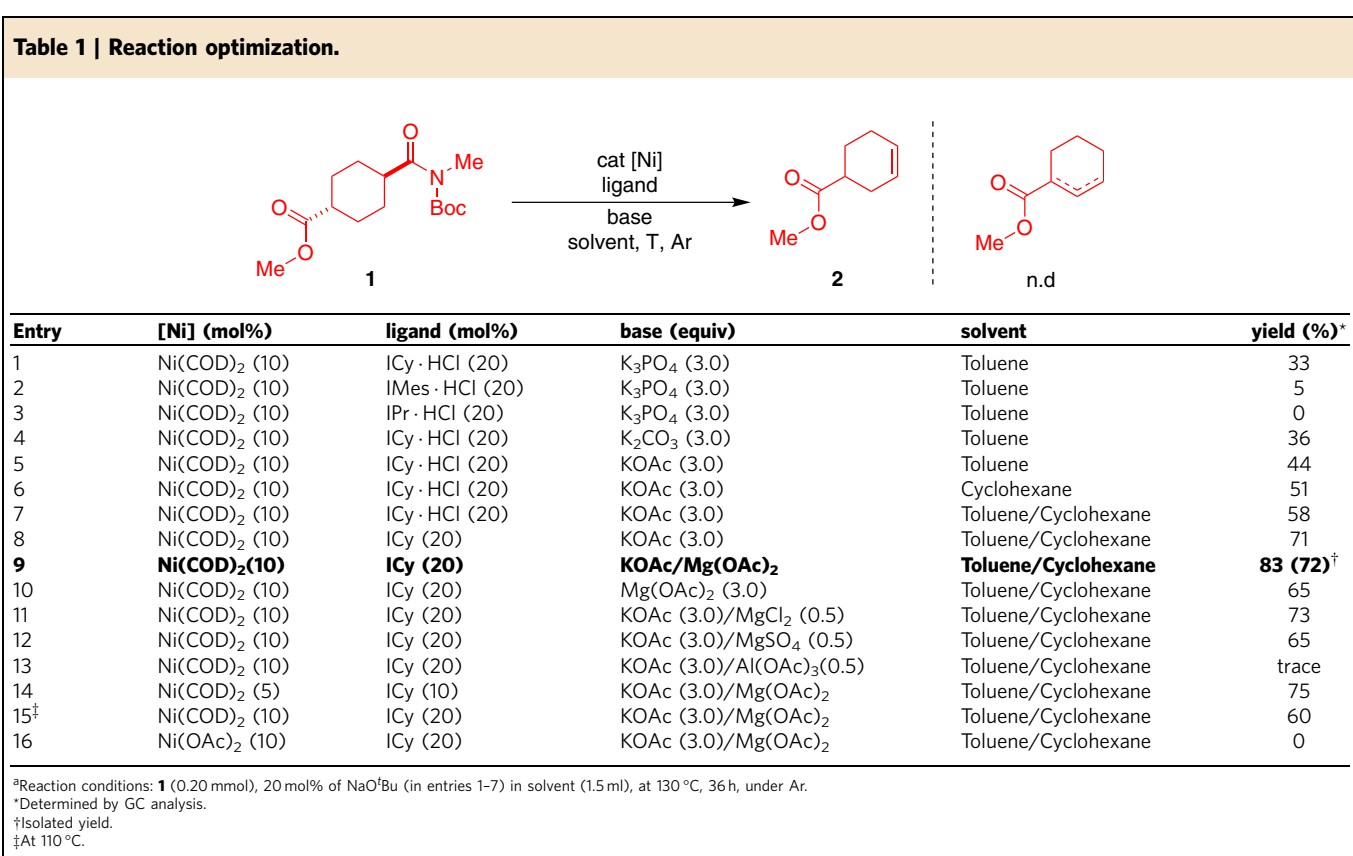

**Figure 2 | Catalytic amide C–N bond activation.** (left cycle) Proposed mechanism on Ni-catalysed decarbonylative borylation of aryl amides. (right cycle) Reaction design for Ni-catalysed decarbonylative elimination of aliphatic amides to olefins.

### Table 1 | Reaction optimization.

| Entry | [Ni] (mol%) | ligand (mol%) | base (equiv) | solvent | yield (%)* |
|---|---|---|---|---|---|
| 1 | Ni(COD)$_2$ (10) | ICy·HCl (20) | K$_3$PO$_4$ (3.0) | Toluene | 33 |
| 2 | Ni(COD)$_2$ (10) | IMes·HCl (20) | K$_3$PO$_4$ (3.0) | Toluene | 5 |
| 3 | Ni(COD)$_2$ (10) | IPr·HCl (20) | K$_3$PO$_4$ (3.0) | Toluene | 0 |
| 4 | Ni(COD)$_2$ (10) | ICy·HCl (20) | K$_2$CO$_3$ (3.0) | Toluene | 36 |
| 5 | Ni(COD)$_2$ (10) | ICy·HCl (20) | KOAc (3.0) | Toluene | 44 |
| 6 | Ni(COD)$_2$ (10) | ICy·HCl (20) | KOAc (3.0) | Cyclohexane | 51 |
| 7 | Ni(COD)$_2$ (10) | ICy·HCl (20) | KOAc (3.0) | Toluene/Cyclohexane | 58 |
| 8 | Ni(COD)$_2$ (10) | ICy (20) | KOAc (3.0) | Toluene/Cyclohexane | 71 |
| **9** | **Ni(COD)$_2$(10)** | **ICy (20)** | **KOAc/Mg(OAc)$_2$** | **Toluene/Cyclohexane** | **83 (72)**[†] |
| 10 | Ni(COD)$_2$ (10) | ICy (20) | Mg(OAc)$_2$ (3.0) | Toluene/Cyclohexane | 65 |
| 11 | Ni(COD)$_2$ (10) | ICy (20) | KOAc (3.0)/MgCl$_2$ (0.5) | Toluene/Cyclohexane | 73 |
| 12 | Ni(COD)$_2$ (10) | ICy (20) | KOAc (3.0)/MgSO$_4$ (0.5) | Toluene/Cyclohexane | 65 |
| 13 | Ni(COD)$_2$ (10) | ICy (20) | KOAc (3.0)/Al(OAc)$_3$(0.5) | Toluene/Cyclohexane | trace |
| 14 | Ni(COD)$_2$ (5) | ICy (10) | KOAc (3.0)/Mg(OAc)$_2$ | Toluene/Cyclohexane | 75 |
| 15[‡] | Ni(COD)$_2$ (10) | ICy (20) | KOAc (3.0)/Mg(OAc)$_2$ | Toluene/Cyclohexane | 60 |
| 16 | Ni(OAc)$_2$ (10) | ICy (20) | KOAc (3.0)/Mg(OAc)$_2$ | Toluene/Cyclohexane | 0 |

[a]Reaction conditions: **1** (0.20 mmol), 20 mol% of NaO$^t$Bu (in entries 1–7) in solvent (1.5 ml), at 130 °C, 36 h, under Ar.
*Determined by GC analysis.
†Isolated yield.
‡At 110 °C.

olefin **2** in 83% yield without isomerization (Table 1, entry 9). Under these conditions, the reaction without KOAc or using other additives had shown inferior results (Table 1, entries 10–13). Notably, a lower loading of Ni(COD)$_2$ can be employed with a slight decreasing in the reaction yield (Table 1, entry 14). Temperature effect was also examined, and decreasing the reaction temperature to 110 °C still resulted in an acceptable yield (Table 1, entry 15). Finally, other nickel sources such as anhydrous Ni(OAc)$_2$ was proven completely unsuccessful in this reaction (Table 1, entry 16).

**Scope of the methodology**. With this exciting initial result in hand, we proceeded to investigate the scope of this reaction (Fig. 3). As shown, we have determined that this Ni-catalysed method can be applied to decarbonylative elimination of a series of aliphatic amides, furnishing the desired internal and terminal olefins in generally good yields. Cyclohexene derivatives with a wide array of functional groups including phenolic ester (**4**), N,N-dimethyl amide (**6**), benzylic ether (**8**) and pentyl group (**10**) can be synthesized in 55–74% yields without isomerizing to other positions. This result is noteworthy because of the known

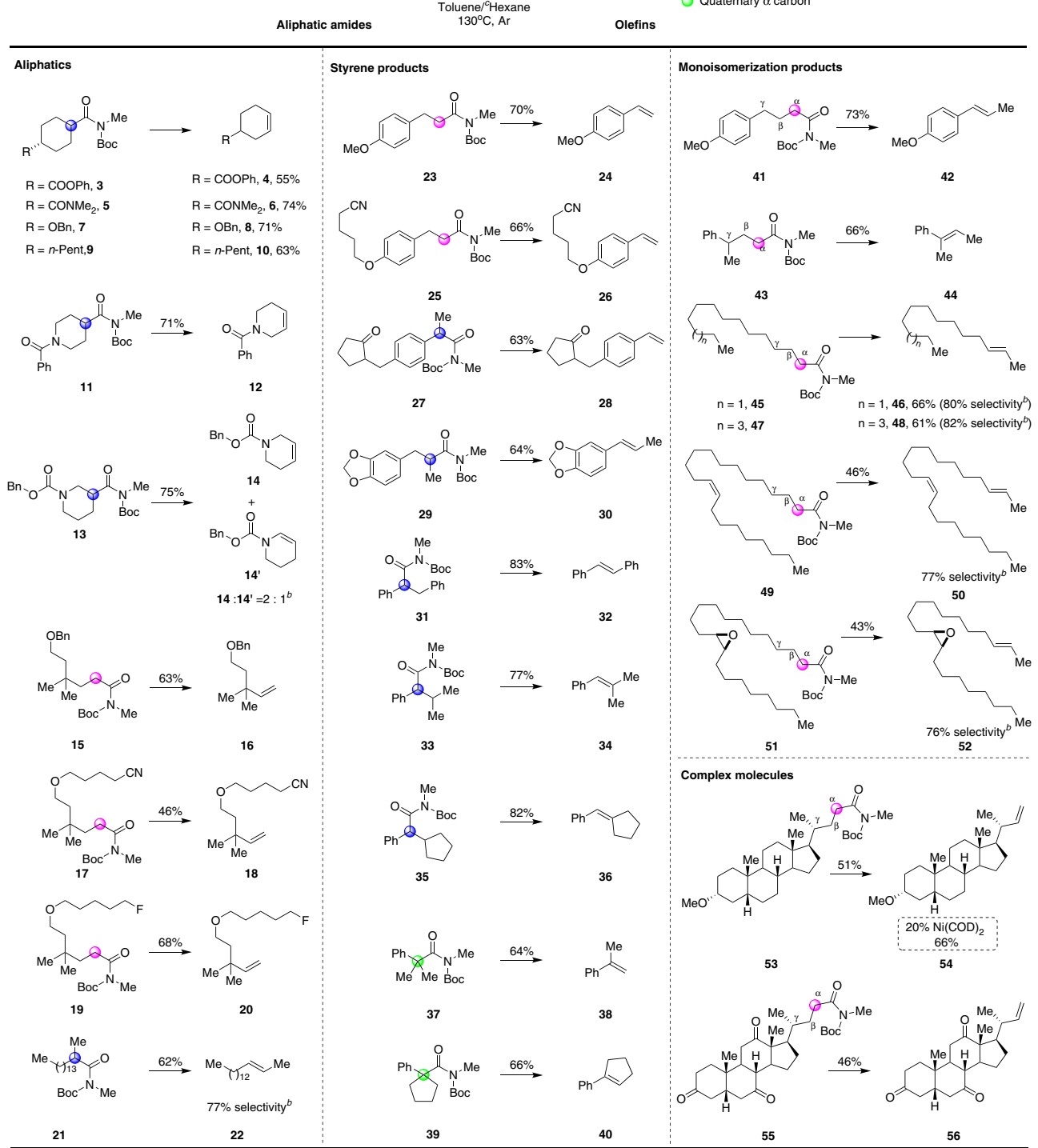

**Figure 3 | Scope of the aliphatic amide retro-hydroamidocarbonylation.** Reaction conditions: amides (0.20 mmol), 10 mol% of Ni(COD)$_2$, 20 mol% of ICy, 0.5 equiv of Mg(OAc)$_2$, 3.0 equiv of KOAc in 3.0 ml toluene/$^c$hexane ($v/v = 1{:}2$) at 130 °C, 36 h, under Ar. $^a$Isolate yield. $^b$Determined by crude $^1$H NMR.

propensity of Ni–H intermediates to catalyse the rapid isomerization of alkenes leading to a mixture of products. *N*-benzoylpiperidine substrate **11** underwent olefination smoothly to provide the corresponding tetrahydropyridine **12** in good yield. Reaction of **13** containing two possible sites for β-hydride elimination gave regioisomeric olefins **14** and **14′** (2:1)

in 75% combined yield, which were separable by column chromatography. Nonactivated, terminal aliphatic amide **15** was also very active substrates in the transformation and provided the desired olefin **16** in 63% yield. The substrate **17** with CN moiety could selectively generate the retro-hydro-amidocarbonylation product **18** and the retrohydrocyanation

**Figure 4 | Synthetic applications of retro-hydroamidocarbonylation reactions.** (**a**) Synthesis of androstadienone (**59**) from etienic acid (**57**). (**b**) Synthesis of the core structure of aspewentin A.

product was not detected under our reaction conditions. The selectivity between the C–F and C–N bond was also investigated with a substrate **19**, in which only C–N cleavage product **20** was observed. A secondary alkyl amide **21** underwent elimination, and produced the internal 2-alkene **22** in 62% yield with 77% selectivity. In addition, a wide range of styrene derivatives could be efficiently prepared by retro-hydroamidocarbonylation of the amides with secondary (**23**, **25**), tertiary (**27**, **29**, **31**, **33**, **35**) and quaternary (**37**,**39**) α carbon centres in good yields. Among them, electronic effect favoured conjugated product **30** over terminal olefin. Elimination of 4-arylbutanoic acid amide **41** and **43** also exhibited the similar effect and generated E-olefins **42** and **44** by monoisomerization of the formed terminal products.

Fatty acids from animals, plants and microorganisms are a unique feedstock for the production of chemicals because of their characteristic long chain methylene sequences. We have transformed saturated fatty acids such namely myristic, and palmitic to corresponding amides and then subjected under the standard protocol to produce 2-alkenes **46** and **48** in 66% and 61% yields respectively with good selectivity. Impressively, unsaturated fatty acids such as erucic acid also participated in the reaction affording diene product **50** in 46% yield. Although it is reported that the strained ring of epoxides was easily opened in nickel catalytic systems, the substrate **51** with an epoxide group was found perfectly tolerable[40]. This nickel-catalysed retro-hydroamidocarbonylation was also viable with complex molecular precursors. The application of the present method resulted in the conversion of amide **53** containing a single γ C–H bond to terminal olefin **54** in 51% yield without isomerization. Moreover, through the use of a higher catalyst loading to 20 mol%, the yield could be improved to 66%. Based on the generated olefin group, 1,4-diol derivative was generated successfully via an iridium-catalysed tandam C–H silylation and oxidation reaction[41]. In case of the derivative of dehydrocholic acid **55** was performed, the corresponding terminal alkene **56**

was also obtained selectively and further isomerization was negligible.

When primary alkyl halides, aldehydes and nitriles were used for transition-metal catalysed elimination, terminal alkenes were obtained as the main products[13,14,17,19]; remarkably, under our conditions, the thermodynamically more stable internal E-2-alkene isomers were mainly obtained from a tandem retro-hydroamidocarbonylation/monoisomerization process in substrates with amide groups adjacent to the secondary carbon centres[42]. It should be noted that the substituents at the γ position also has a large effect on this process. The aryl group at the γ position favours monoisomerization while the alkyl group inhibits the isomerization (**44** versus **54** and **56**).

**Synthetic applications**. To fully demonstrate the applicability of this methodology with predictable rules, we prepared a steroid product androstadienone (**59**), which was described to exhibit potent pheromone-like activities in human (Fig. 4)[43]. The synthesis commenced with the preparation of amide **58** from commercially available inexpensive etienic acid **57**. By using retro-hydroamido-carbonylation as a key step, decarbonylative elimination of **58** afforded the costly androstadienone (**59**) in 43% yield without olefin-isomerization according to the rules outlined above. The formed double bond can be further functionalized to ketone to construct another compound Trichiliasterone B[44]. Since many methods have been developed to access N-Boc-N-Me amide groups, it allows facile application of the retro-hydroamido-carbonylation reaction to the laboratory-scale synthesis of nature products. A convincing example is using acrylamide **61** as a building block in construction of nature product skeleton. Starting from olefin **60** and acrylamide **61**, we constructed substrate **62** via a reductive olefin coupling[45]. The decarbonylative elimination process could be applied to synthesize the key intermediate **63** in 70% yield as a core structure of aspewentin A, a norditerpene natural product isolated from Aspergillus wentii[46,47].

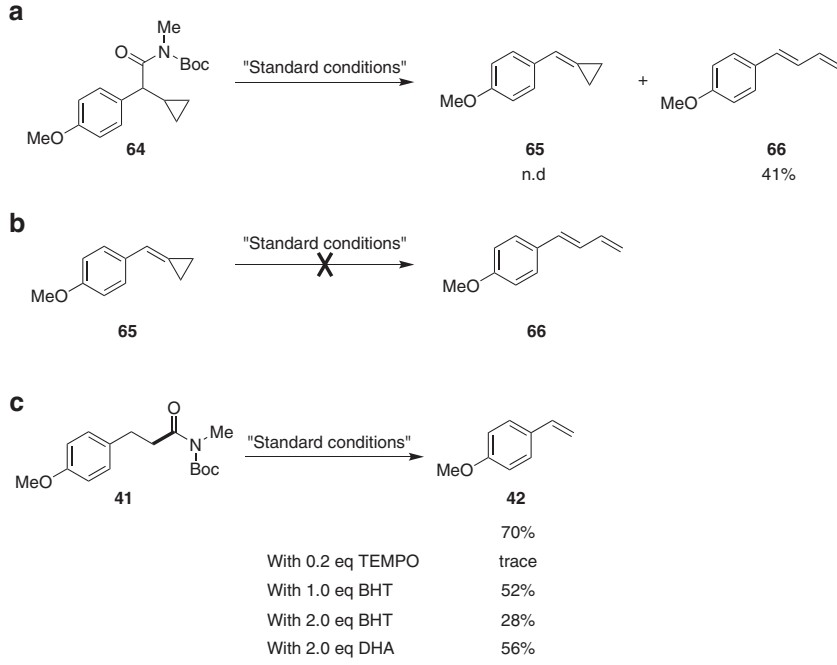

**Figure 5 | Investigation for mechanistic insights. (a)** Reaction of a radical clock substrate. **(b)** Exclusion of the conversion from compounds **65** to **66**. **(c)** Reaction in the presence of radical inhibitors.

## Discussion

Several experiments were also conducted to provide insight about the potential mechanism of this transformation. As shown in Fig. 5a, radical clock substrate **64** generated the ring-open product **66** in 41% yield under the standard conditions, without detection of cyclopropane-containing product **65**. Moreover, using the synthesized compound **65**, it could not generate the diene product **66** in this Ni-catalysed system (Fig. 5b). These results demonstrated that this reaction proceed through a radical process. In addition, radical inhibiting experiments were carried out in the presence of radical traps (Fig. 5c). The reaction was largely inhibited when 0.2 equiv. TEMPO (the radical inhibitor TEMPO led to poor conversion, also presumably from catalyst inhibition[48]) or 2.0 equivalent BHT and DHA was added, also indicating a radical type reaction mechanism. This finding is inconsistent with the pathway we originally envisaged in Fig. 2. Because the C–N bond activation and decarbonylative process have been confirmed in our previous work[30,31], we suspected that the formed intermediate **C′** in aliphatic amides could occur homolytic cleavage reversibly[49]. Nonetheless, details for the mechanism of this reaction are not clear at present and more studies are required to fully elucidate the reaction mechanism.

In summary, we have developed an efficient Ni-catalysed system, which is capable of activating aliphatic amide C–N bonds for decarbonylative elimination to produce various olefins. (During the revision of this manuscript, Garg and coworkers reported Nickel-catalyzed C-N cleavage of aliphatic amides to construct esters[50].) The application of this method to synthesize complex unsaturated molecules has also been illustrated. In view of the widespread utility of amide group, this method offers a very meaningful tool in synthetic transformation. Studies to determine further mechanistic details of this retro-hydroamidocarbonylation process and to expand the scope of aliphatic amide C–N bond activation to other transformations are underway in our laboratory.

## Methods

**General procedure (2).** To a 25 ml Schlenk flask equipped with a magnetic stir bar was charged with 0.2 mmol 4-((tert-butoxycarbonyl)(methyl)carb-amoyl)cy-clohexane-1-carboxylate (**1**). The tube was introduced in nitrogen-filled glovebox, and Ni(COD)$_2$ (5.5 mg, 10 mol %), ICy (9.3 mg, 20 mol %), KOAc (58.9 mg, 3.0 equiv), Mg(OAc)$_2$ (14.2 mg, 0.5 equiv) were added. The tube with the mixture was taken out of the glovebox. Then cyclohexane (1.0 ml) and toluene (0.5 ml) were added under argon. The formed mixture was stirred at 130 °C under Ar for 36 h as monitored by TLC and GC-MS. The solution was then cooled to room temperature. The crude product was further purified by column chromatography on silica gel (*n*-pentane:AcOEt = 50:1) to afford 20.2 mg (72%) of methyl cyclohex-3-ene-1-carboxylate (**2**) as a colourless oil. For NMR spectra of the compounds in this manuscript, see Supplementary Figs 1–127. For details of the synthetic procedures, see Supplementary Methods.

**Data availability.** The authors declare that the data supporting the findings of this study are available within the article and its Supplementary Information Files.

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

## Acknowledgements

We thank the '1000-Youth Talents Plan', the 'Jiangsu Specially-Appointed Professor Plan', NSF of China (Grant 21402086, 21672097) and NSF of Jiangsu Province (Grant BK20140594) for financial support. This work was also supported by a Supported by the program for Outstanding PhD candidate of Nanjing University.

## Author contributions

J.H. and X.P. performed the experiments. Z.S. conceived the concept, directed the project and wrote the paper. M.W. and Z.S. discussed the results and commented on the manuscript.

## Additional information

**Competing interests:** The authors declare no competing financial interests.

**Publisher's note**: 

