## [Peer review file · Nature Communications]

Reviewers' comments:

Reviewer #1 (Remarks to the Author):

In the submitted manuscript, Shi and co-workers presented a new method to perform decarbonylative elimination reaction of aliphatic amides to olefins using Ni/NHC-catalytic system which could be recognized as the reverse reaction of hydroamidocarbonylation. This work was based on the nickel-catalyzed decarbonylative cross-coupling reactions developed by the authors' research group (ref. 30, 31) as well as inspired by the findings of Garg and Houk's group (ref 3-6).

The chemistry showed the broad substrate scope with a range of functional groups (Figure 3). The successful application to complex molecules furthermore increased the synthetic level of this protocol (Figure 4). Although the activation and further transformation of amide C-N has been realized, it was limited to aryl and hetero-aryl amide substrates. This work presented an interesting and useful example for the C-N & C-C cleavage and further conversion of aliphatic amides.

The novelty and synthetic potential of this protocol should be of great interest to audience in synthetic communities as well as more general scientific field. Thus, I do think this paper meets the criteria for publication in Nat. Commun.

Nickel is less electronegative than Pd (Nature 2014, 509, 299), makes it much easier for nickel catalysts to activate less reactive electrophiles such as C-N bond of amides. However, the beta-hydride elimination with nickel was much slower or even thermodynamically disfavored. The combination of activation of less reactive electrophiles with beta-hydride elimination might not be the only best result for Nickel catalysts. It might be possible to use some special nucleophiles to carry out the cross-coupling of aliphatic amides with a similar decarbonylation process.

Several synthetically relevant functional groups should be tested, such as fluoride, nitrile, aldehyde and epoxide group. The substrates with fluoride atom could give out the selectivity information between the C-F and C-N. The substrates with nitrile or aldehyde group could be converted to olefins according to the cited previous literatures (ref 17-19). The strained ring of epoxides were easily opened in nickel catalytic systems.

Reviewer #2 (Remarks to the Author):

The authors herein describe the development of an interesting retro-hydroamido carbonylation reaction of aliphatic amides to olefins by Ni/NHC catalysis. The reported reaction is inspired by the previous work of retro-hydroformylation by Dong and Nozaki, retro-hydrocyanation by Morandi, as well as the recent progress on nickel-catalyzed acyl C-N bond activation by Garg and Szostak. In this work, various aliphatic amides with secondary, tertiary and quaternary α -carbons can all undergo the present reaction condition smoothly to afford internal and terminal olefins with good regioselectivity and stereoselectivity. The scope is reasonable and this methodology is well applied to complex molecules synthesis. In general this reaction represents a good advance in metal-catalyzed transformations and will be interesting to the synthetic communities.

However, the proposed mechanism is vague and one would expect a more detailed discussion supported by experiments or/and DFT calculations.

Further minor revisions have to be addressed:

1. For substrate 3 with a phenolic ester group in Figure 3, the yield of corresponding product 4 is only 55%. What is the side reaction? Did the authors observe the formation of side products that come from the decarbonylative elimination of ester?
2. For compound 18, does the '77% selectivity' mean that only E-isomer was afforded and the ratio of internal 2-alkene and terminal 1-alkene was 77:23? I suggest the authors should write it more clearly, just like 14 and 14'. To the product containing possible Z&E-isomers, the authors should clearly show the Z&E ratio in the scope figure.
3. In the Supporting Information, the NMR peak size in some of the spectra is reduced too much, even less than 20% (eg. 7, 9, 2, 4, 20, 24, 26, 38, 40, 48, 53). And several compounds contain impurities that appear to be more than 5% (eg. 4, 20, 27).
4. What is the role of base in the reaction mechanism? The authors observed the desired product in 65% yield in the presence of magnesium acetate. What is the result for the reaction in the absence of base and additive?
5. According to the proposed reaction mechanism, did the authors observe H-NMeBoc as byproduct?

In summary the manuscript is well written and conducted with care. However, several issues have to be addressed before publication in Nature Communications can be recommended.

Reviewer #1:

In the submitted manuscript, Shi and co-workers presented a new method to perform decarbonylative elimination reaction of aliphatic amides to olefins using Ni/NHC-catalytic system which could be recognized as the reverse reaction of hydroamidocarbonylation. This work was based on the nickel-catalyzed decarbonylative cross-coupling reactions developed by the authors' research group (ref. 30, 31) as well as inspired by the findings of Garg and Houk's group (ref 3-6).

The chemistry showed the broad substrate scope with a range of functional groups (Figure 3). The successful application to complex molecules furthermore increased the synthetic level of this protocol (Figure 4). Although the activation and further transformation of amide C-N has been realized, it was limited to aryl and hetero-aryl amide substrates. This work presented an interesting and useful example for the C-N & C-C cleavage and further conversion of aliphatic amides.

The novelty and synthetic potential of this protocol should be of great interest to audience in synthetic communities as well as more general scientific field. Thus, I do think this paper meets the criteria for publication in Nat. Commun.

Nickel is less electronegative than Pd (Nature 2014, 509, 299), makes it much easier for nickel catalysts to active less reactive electrophiles such as C-N bond of amides. However, the beta-hydride elimination with nickel was much slower or even thermodynamically disfavored. The combination of activation of less reactive electrophiles with beta-hydride elimination might not be the only best result for Nickel catalysts. It might be possible to use some special nucleophiles to carry out the cross-coupling of aliphatic amides with a similar decarbonylation process.

Answer: Thanks for this kind suggestion! The utilization of some interesting nucleophiles in aliphatic amide C-N bond activation is underway in our laboratory.

Several synthetically relevant functional groups should be tested, such as fluoride, nitrile, aldehyde and epoxide group. The substrates with fluoride atom could give out the selectivity information between the C-F and C-N. The substrates with nitrile or aldehyde group could be

converted to olefins according to the cited previous literatures (ref 17-19). The strained ring of epoxides were easily opened in nickel catalytic systems.

Answer: Thanks for this suggestion! We tried several substrates with fluoride, nitrile and epoxide groups as shown below. To our delight, these groups were found perfectly tolerable in our catalytic system. The new results have been added in the revised manuscript (see Figure 3, “**17, 19, 25, 51**”). Thanks again.

Reviewer #2:

The authors herein describe the development of an interesting retro-hydroamido carbonylation reaction of aliphatic amides to olefins by Ni/NHC catalysis. The reported reaction is inspired by the previous work of retro-hydroformylation by Dong and Nozaki, retro-hydrocyanation by Morandi, as well as the recent progress on nickel-catalyzed acyl C-N bond activation by Garg and Szostak. In this work, various aliphatic amides with secondary, tertiary and quaternary α -carbons can all undergo the present reaction condition smoothly to afford internal and terminal olefins with good regioselectivity and stereoselectivity. The scope is reasonable and this methodology is well applied to complex molecules synthesis. In general this reaction represents a

good advance in metal-catalyzed transformations and will be interesting to the synthetic communities.

However, the proposed mechanism is vague and one would expect a more detailed discussion supported by experiments or/and DFT calculations.

Answer: We did some additional experiments to provide more detailed insight about the potential mechanism of this transformation. Radical clock substrate **64** generated the ring-open product **66** in 41% yield under the standard conditions, without detection of cyclopropane-containing product **65** (a). Moreover, using the synthesized compound **65**, it could not generate the diene product **66** in this Ni-catalyzed system (b). These results demonstrated that this reaction proceed through a radical process. In addition, radical inhibiting experiments using TEMPO, BHT and DHA as radical traps were carried out (c). The reaction was largely inhibited when 0.2 equiv. TEMPO or 2.0 equivalent BHT and DHA was added, also indicating a radical type reaction mechanism. This finding is inconsistent with the pathway we originally envisaged in Figure 2. Because the C-N bond cleavage and decarbonylative process have been confirmed in our previous work, we suspected that the formed intermediate **C'** in aliphatic amides could occur homolytic cleavage reversibly. Nonetheless, details for the mechanism of this reaction are not clear at present. Further investigations are ongoing in our lab.

Further minor revisions have to be addressed:

1. For substrate 3 with a phenolic ester group in Figure 3, the yield of corresponding product 4 is only 55%. What is the side reaction? Did the authors observe the formation of side products that come from the decarbonylative elimination of ester?

Answer: We didn't observe the product on decarbonylative elimination of ester group. The lower yield is resulted from the hydrolysis of the phenolic ester group in starting material. This undesired carboxylic acid can be detected in GC-MS and crude H NMR (around 30%).

2. For compound 18, does the '77% selectivity' mean that only E-isomer was afforded and the ratio of internal 2-alkene and terminal 1-alkene was 77:23? I suggest the authors should write it more clearly, just like 14 and 14'. To the product containing possible Z&E-isomers, the authors should clearly show the Z&E ratio in the scope figure.

Answer: The '77% selectivity' mean the only E-isomer. We didn't observe the Z isomers in our reaction. The '23% selectivity' includes 1-alkene and other internal alkenes. These isomerization products are difficult to be distinguished one by one from H NMR.

3. In the Supporting Information, the NMR peak size in some of the spectra is reduced too much, even less than 20% (eg. 7, 9, 2, 4, 20, 24, 26, 38, 40, 48, 53). And several compounds contain impurities that appear to be more than 5% (eg. 4, 20, 27).

Answer: Thanks for this kind reminding! We have revised the SI according to your suggestions.

4. What is the role of base in the reaction mechanism? The authors observed the desired product in 65% yield in the presence of magnesium acetate. What is the result for the reaction in the absence of base and additive?

Answer: According to the proposed reaction mechanism, one equivalent HX was formed during the reaction and the base could speed up reactions. Without the additional base, we can also get the olefination product in 48%. We proposed that Mg(OAc)₂ could also act as a Lewis acid coordination with two oxygen atom from amide and Boc group to activate the C-N bond.

5. According to the proposed reaction mechanism, did the authors observe H-NMeBoc as byproduct?

Answer: Yes. The H-NMeBoc can be detected in GC-MS and crude H NMR.

In summary the manuscript is well written and conducted with care. However, several issues have to be addressed before publication in Nature Communications can be recommended.

REVIEWERS' COMMENTS:

Reviewer #1 (Remarks to the Author):

The authors have well addressed my concerns. I am happy to recommend publication of the revised manuscript.

Reviewer #2 (Remarks to the Author):

I am happy with the Revision and recommend publication of this manuscript

REVIEWERS' COMMENTS:

Reviewer #1 (Remarks to the Author):

The authors have well addressed my concerns. I am happy to recommend publication of the revised manuscript.

Answer: We sincerely appreciate the reviewer for the very positive recommendation on the manuscript.

Reviewer #2 (Remarks to the Author):

I am happy with the Revision and recommend publication of this manuscript

Answer: We sincerely appreciate the reviewer for the very positive recommendation on the manuscript.